# Elevated Plasma Levels of Circulating Extracellular miR-320a-3p in Patients with Paroxysmal Atrial Fibrillation

**DOI:** 10.3390/ijms21103485

**Published:** 2020-05-15

**Authors:** Andrey V. Zhelankin, Sergey V. Vasiliev, Daria A. Stonogina, Konstantin A. Babalyan, Elena I. Sharova, Yurii V. Doludin, Dmitry Y. Shchekochikhin, Eduard V. Generozov, Anna S. Akselrod

**Affiliations:** 1Department of Molecular Biology and Genetics, Federal Research and Clinical Center of Physical-Chemical Medicine of Federal Medical Biological Agency, 119435 Moscow, Russia; babalyan@rcpcm.org (K.A.B.); sharova.ei@rcpcm.org (E.I.S.); generozov@rcpcm.org (E.V.G.); 2Department of Cardiology, Functional and Ultrasound Diagnostics, Faculty of Medicine N.V. Sklifosovsky, I.M. Sechenov First Moscow State Medical University of the Ministry of Health of the Russian Federation (Sechenov University), 119146 Moscow, Russia; lod.kain@gmail.com (S.V.V.); stonogina.d@gmail.com (D.A.S.); agishm@list.ru (D.Y.S.); 7402898@mail.ru (A.S.A.); 3FSI National Research Center for Preventive Medicine of the Ministry of Health of the Russian Federation, 101990 Moscow, Russia; ydoludin@gnicpm.ru

**Keywords:** circulating microRNA, miR-320a, paroxysmal atrial fibrillation, non-invasive biomarker

## Abstract

The potential of extracellular circulating microRNAs (miRNAs) as non-invasive biomarkers of atrial fibrillation (AF) has been confirmed by a number of recent studies. However, the current data for some miRNAs are controversial and inconsistent, probably due to pre-analytical and methodological differences. In this work, we attempted to fulfill the basic pre-analytical requirements provided for circulating miRNA studies for application to paroxysmal atrial fibrillation (PAF) research. We used quantitative PCR (qPCR) to determine the relative plasma levels of circulating miRNAs expressed in the heart or associated with atrial remodeling or fibrillation with reported altered plasma/serum levels in AF: miR-146a-5p, miR-150-5p, miR-19a-3p, miR-21-5p, miR-29b-3p, miR-320a-3p, miR-328-3p, miR-375-3p, and miR-409-3p. First, in a cohort of 90 adult outpatient clinic patients, we found that the plasma level of miR-320a-3p was elevated in PAF patients compared to healthy controls and hypertensive patients without AF. We further analyzed the impact of medication therapies on miRNA relative levels and found elevated miR-320a-3p levels in patients receiving angiotensin-converting-enzyme inhibitors (ACEI) therapy. Additionally, we found that miR-320a-3p, miR-21-5p, and miR-146a-5p plasma levels positively correlated with the CHA_2_DS_2_-Vasc score and were elevated in subjects with CHA_2_DS_2_-Vasc ≥ 2. Our results indicate that, amongst the analyzed miRNAs, miR-320a-3p may be considered as a potential PAF circulating plasma biomarker, leading to speculation as to whether this miRNA is a marker of platelet state change due to ACEI therapy.

## 1. Introduction

New biomarkers of atrial fibrillation (AF), the most common arrhythmia with a high risk of morbidity and mortality, may improve the clinical risk assessment and understanding of the pathophysiology of this condition [1]. MicroRNAs (miRNAs) are a class of small RNAs that are extraordinarily stable in biofluids and are considered to be promising non-invasive biomarkers for many pathological processes. Change in the profile of circulating blood extracellular miRNAs has been detected in a large number of human pathologies, including cardiovascular diseases (CVD) [2,3,4,5]. A number of studies have shown that AF is accompanied by altered circulating miRNA levels [6]. AF studies in patients and animal models suggest that both the electrical and structural remodeling of atria contribute to the occurrence and maintenance of AF [7]. The remodeling of atrial myocytes, involving structural, metabolic, and electrophysiological changes, is an essential component of AF pathogenesis [8]. The miRNA expression profile of atrial myocytes in AF is changed in comparison to unaffected atria [9]. The potential mechanism of changing the profile of extracellular circulating plasma miRNAs in AF is the secretion of exosomes by atrial cardiomyocytes. These exosomes contain a specific miRNA signature altered in AF [10]. Recent research suggests that plasma exosomes from AF patients are enriched with some miRNA species compared to those of individuals with a sinus rhythm [11]. However, the secretion of miRNA-containing components by the vascular endothelial and smooth muscle cells, blood cells, and cells of any tissues that are in contact with the bloodstream also contributes to the general profile of blood plasma miRNAs. Each type of blood peripheral cell contains a unique miRNA profile [12]. Platelets, as well as platelet-derived microparticles (PMPs), also contain miRNAs [13,14]. The total circulating plasma or serum miRNA is a mixture of miRNAs originating from exosomes, microvesicles, apoptotic bodies, and miRNA-protein and miRNA-HDL (high density lipoprotein) complexes [15]. Obtaining a cell-free and platelet-poor or platelet-free plasma (PPP or PFP, respectively) by blood centrifugation is important in any study of circulating extracellular miRNAs [16,17]. However, platelet elimination by centrifugation does not mean the elimination of PMPs, which are likely to be the main source of circulating miRNAs in plasma or serum [18,19]. The levels of both platelet and serum miR-150 were lower in the presence of AF in patients with heart failure [20]. The circulating level of some platelet-enriched miRNAs is increased in response to acute platelet activation, and changes in the platelet-derived miRNA pool may occur due to antiplatelet therapy [18,21]. Platelets and PMPs contain a number of plasma miRNAs dysregulated in AF and AF-related diseases (miR-146a, miR-21, miR-150, miR-19a, and miR-320a) [6,18,22,23,24,25]. Therefore, it should be taken into account that changes in the plasma miRNA profile in AF patients may be PMP-driven and reflect a platelet functional state. Red blood cells (RBC) also contain miRNAs. Therefore, the assessment of RBC hemolysis in plasma/serum samples is an important and highly recommended pre-analytical step in the circulating miRNA studies [26,27]. Hemolysis affects plasma levels of some AF-associated miRNAs, such as miR-19a-3p, miR-21-5p, miR-320a-3p, and miR-16-5p—a widely used endogenous normalization control in miRNA studies [28].

In this study, we analyzed the plasma levels of circulating miRNAs in patients with paroxysmal AF (PAF) compared to healthy controls and controls with arterial hypertension (HT). The analysis included miRNA species expressed in the heart or associated with atrial remodeling or fibrillation with reported plasma/serum levels altered in AF. The current data for some miRNAs are controversial and inconsistent between different studies [6]. Given the significant influence of pre-analytical factors on the blood circulating extracellular miRNA content, these discrepancies are probably caused by the initial choice of biofluid (plasma or serum), different plasma/serum preparation protocols, a lack of accurate control of sample hemolysis, different techniques for the isolation and analysis of miRNAs, and the use of different normalization controls for relative miRNA level measurements [29,30,31,32,33]. In addition, medication therapies affecting the platelet state may alter the miRNA profile in each particular sample. In the current study, we attempted to fulfill the basic pre-analytical requirements provided for circulating miRNA studies applied to PAF research.

## 2. Results

### 2.1. Study Sample Characteristics

A total of 90 subjects were included in this study, according to clinical inclusion/exclusion criteria and plasma/miRNA sample quality requirements. The study sample included three groups:PAF: 30 PAF patients;HT: 30 hypertensive patients without AF;CONTR: 30 healthy controls.

The main characteristics of the study sample groups are shown in Table 1. Age gaps between HT patients and controls and between PAF and HT patients were ~10 years, which is consistent with the global data on the age-specific prevalence and incidence of AF [7,34]. The PAF group included five cases of type 2 diabetes mellitus (DM) and seven cases of stable coronary artery disease (CAD). The left atrial volume was ~1.5-fold higher in PAF patients compared to controls and HT patients (Mann–Whitney, *p* < 0.05). Distribution plots of the main characteristics in the study groups are shown in Appendix A. Mann–Whitney *p*-values of significance for comparisons of these characteristics with those of groups with Bonferroni–Holm correction are given in Appendix A.

### 2.2. Spectrophotometric and miRNA-qPCR-Based Hemolysis Evaluation

The results of the spectrophotometric and miRNA-qPCR-based hemolysis assessment of six individual plasma samples with different hemolysis degrees are shown in Figure 1. The difference between qPCR quantitative cycles (Cq) of miR-23a-3p and miR-451a (dCq _(miR-23a-3p–miR-451a)_ value), used for qPCR-based hemolysis assessment, tended to increase steadily with the increase of the hemolysis degree measured by spectrophotometry. In the original study, which used the LNA™-based qPCR method to obtain Cq values of miR-451a and miR-23a-3p for hemolysis assessment, the authors considered a dCq _(miR-23a-3p–miR-451a)_ of 7–8 or more as an indicator of a high hemolysis risk [19]. Our qPCR data were obtained with different technology (TaqMan Advanced by Thermo Fisher Scientific) and we observed different Cq and dCq values for miR-451a and miR-23a-3p. We observed dCq _(miR-23a-3p–miR-451a)_ values of 9–11, which corresponded to the samples with the lowest hemolysis degree. Values of 14–16 corresponded to hemolyzed samples, which is equivalent to a ~16,000–65,000-fold difference between plasma levels of these miRNAs. Samples with dCq _(miR-23a-3p–miR-451a)_ > 14 were not included in this study due to the possible influence of hemolysis on miRNA plasma levels.

The results of the hemolysis assessment of plasma samples from the study groups are shown in Table 2. The dCq _(miR-23a-3p–miR-451a)_ values in the studied samples ranged from 7.9 to 13.5. Appierto et al. [35] used a hemolysis score (HS) threshold of 0.057 to distinguish hemolyzed and non-hemolyzed plasma samples. In our study, HS values ranged from 0.01 to 0.25 (median 0.116), and only ~6% of samples had HS < 0.057. There were no significant differences in the values of any spectrophotometric-based hemolysis indices between the study groups. Among these indices, HS showed the highest association with dCq _(miR-23a-3p–miR-451a)_ in the whole sample (adjusted linear R^2^ = 0.243, *p* = 4.876 × 10^−7^); despite the correlation between these two values (Spearman’s Rho 0.504, *p* < 0.05), HS did not reflect the miRNA ratio to a sufficient degree (Figure 2). Both HS and dCq _(miR-23a-3p–miR-451a)_ values were used as confounding factors in multiple linear regression (MLR) analysis.

### 2.3. Plasma miRNA Levels

The MLR analysis showed that three miRNAs had significantly altered levels in PAF patients compared to healthy controls (Table 3). Among those miRNAs, the elevation of miR-320a-3p in PAF patients was the most pronounced factor based on log2(fold change) values, and miR-320a-3p plasma levels were also higher in PAF compared to HT patients. The distribution of relative miRNA plasma levels in the analyzed groups is shown in Figure 3.

To estimate the presence of a combination effect of different miRNAs, we performed MLR analysis checking the interaction between at least two miRNAs for group comparisons, using age and hemolysis indices as confounding factors. As a result, we did not find any statistically significant interactions in the PAF vs. CONTR, PAF vs. HT, and PAF vs. CONTR+HT comparisons.

To analyze the association between relative miRNA plasma levels and the CHA_2_DS_2_-Vasc score, the study sample was divided into two groups: subjects with CHA_2_DS_2_-Vasc < 2 (N = 48) and subjects with CHA_2_DS_2_-Vasc ≥ 2 (N = 42). The MLR analysis showed that three miRNAs (miR-320a-3p, miR-21-5p, and miR-146a-5p) had significantly altered levels, with log2(fold change) > 1.5 in subjects with CHA_2_DS_2_-Vasc ≥ 2 (Table 4). These miRNAs also showed the most pronounced correlation between their relative plasma levels and the CHA_2_DS_2_-Vasc score (Spearman’s Rho > 0.5, *p* < 1.00 × 10^−9^, Table 4).

To estimate the relationship between miRNA plasma levels and any medication therapies received by patients, we performed MLR analysis for each miRNA and types of potentially influencing therapies in a combined PAF+HT group (N = 60), using age and hemolysis indices as confounding factors. The results of this analysis showed a statistically significant relationship between miR-320a-3p, miR-146a-5p, and miR-21-5p plasma levels and angiotensin-converting enzyme inhibitors (ACEI) therapy (Appendix A).

Amongst these miRNAs, miR-320a-3p was of greater interest as it represented a significant increase in the plasma of PAF patients compared to hypertensive patients, i.e., between groups receiving medical therapies. In our cohort, only 10% of PAF patients had antiplatelet therapy, so there was no evidence to suggest that elevated levels of miR-320a-3p in PAF were a response to therapy-mediated inhibition of platelet activation. To estimate whether the elevation of miR-320a-3p plasma levels in PAF patients was the result of other medication therapies potentially influencing the platelet state, we compared miRNA levels of PAF patients with and without taking beta-blockers, calcium channel blockers, ACEI, diuretics, anticoagulants, and statins. There were no significant differences in the miR-320a-3p level between any of the compared groups (Mann–Whitney test).

We also performed the same analysis within the HT group and observed a statistically significant increase in miR-320a-3p levels in HT patients receiving ACEI compared to those not receiving ACEI treatment (*p* = 0.039, Mann–Whitney test). In a combined PAF+HT group, we also observed a statistically significant increase in miR-320a-3p levels in patients receiving ACEI (*p* = 0.014, Mann–Whitney test). The plots of distribution of relative miR-320a-3p plasma levels in patients with and without ACEI treatment in HT and PAF groups are presented in Figure 4.

## 3. Discussion

In our study, we first performed an analysis of plasma miRNAs in paroxysmal atrial fibrillation with a detailed evaluation of the main pre-analytical parameters required for the correct measurement of circulating miRNA biomarkers. Using hemolysis indices and the presence of concomitant diseases as confounding factors in statistical analysis, we observed a moderate increase in relative plasma levels of circulating hsa-miR-320a-3p in patients with PAF compared to healthy controls and hypertensive patients without AF.

Compliance with a strict and standardized protocol for plasma preparation with the elimination of any cellular components—nuclear cells, platelets, erythrocytes, and cellular debris—is crucial in studies on circulating extracellular miRNAs. In this study, we used two-step centrifugation for PFP preparation, according to Duttagupta et al. [17]. A single additional centrifugation step minimizes the level of contaminating cellular RNA in the plasma sample, preserving the expression of circulating miRNA species [17]. Another important pre-analytical problem is bias due to the effect of RBC hemolysis on circulating miRNA levels. In our study, this problem was of key importance, since we used miR-16-5p as a reference endogenous control for miRNA plasma level normalization. Plasma miR-16 levels show small differences between individuals with different physiological conditions, but are significantly affected by the presence of RBC hemolysis [27,28]. A number of AF-associated microRNAs analyzed in this study (miR-19a-3p, miR-21-5p, and miR-320a-3p) are also sensitive to hemolysis [27,28]. Therefore, accurate hemolysis control is a way to reduce the possible bias in miRNA detection. The spectrophotometric and qPCR-based hemolysis assessment was performed for all plasma samples included in this study. We found that the spectrophotometric-based HS index correlates with the hemolysis-dependent miR-23a-3p/miR-451a miRNA ratio, and established the criteria for plasma sample inclusion: HS < 0.25 and dCq _(miR-23a-3p–miR-451a)_ < 14.

Amongst the 10 target miRNAs chosen for analysis in this study, one miRNA (hsa-miR-432-5p) had extremely low Cq values or was undetectable in the test experiments and was excluded from the study sample analysis. Mean Cq values of other miRNAs ranged from ~12 to ~31, which is equivalent to a ~500,000-fold difference between plasma levels of the highest-expressed (miR-451a) and lowest-expressed (miR-409-3p) miRNAs. To evaluate the consistency of our data with the present plasma whole miRNA profile studies, we compared the relative plasma levels of target miRNAs with those obtained from the two most recent publications which used plasma small RNA sequencing in a cohort of healthy subjects (Max et al., 2018 [36] and Godoy et al., 2018 [37]). These studies contained the miRNA sequencing (miRNA-seq) data obtained from 13 and 12 adult plasma samples, respectively. For a comparison with our qPCR data, relative plasma levels from sequencing data were calculated as normalized read counts of each miRNA related to read counts of miR-16-5p. The results of this comparison are presented in Figure 5. The direct comparison of miRNA-seq data with qPCR data is questionable due to the different workflow and data analysis procedures. Moreover, even an analysis of the same samples with different miRNA-seq approaches may give highly divergent results [38]. In addition, the data obtained in different study populations are affected by pre-analytical factors, age, the presence of concomitant diseases, and the use of medication therapies. However, such comparisons may be useful for any further qPCR validation of miRNA-seq results. The concordance of the results between two sequencing studies for the analyzed miRNAs was higher than between our results and any of these studies. Though the direction of change for most miRNAs was similar, our study showed significantly lowered miR-16-5p-normalized relative plasma levels for the majority of miRNAs compared to miRNA-seq data. Low or undetectable plasma levels of miR-432-5p were in accordance with plasma miRNA-seq studies.

In this study, we did not observe notably altered relative plasma miRNA levels in PAF patients compared to controls for the majority of analyzed miRNAs previously reported as AF-associated. This could be explained by both the different miRNA analysis methodologies employed and clinical differences in the studied groups. Unlike previous studies with a similar design, we used relatively novel TaqMan Advanced miRNA Assays with universal reverse transcription, which is more convenient in the simultaneous analysis of dozens of miRNA targets than single TaqMan assays. Additionally, we used detailed hemolysis control for plasma sample inclusion and hemolysis-related criteria in data analysis, whereas the majority of previous studies did not provide data on the hemolysis assessment. Our study sample included patients with a paroxysmal form of AF, which constitutes approximately half of all AF cases and is thought to represent an early stage of the disease [39]. miRNA profile changes for patients with this initial AF diagnosis may not have such pronounced changes compared to more severe AF forms.

Our main findings concern elevated miR-320a-3p as a potential plasma marker for PAF patients. Sommariva et al. [22] previously reported lowered plasma levels of miR-320a in patients with arrhythmogenic cardiomyopathy (ACM) compared to healthy controls using qPCR and endogenous plasma miR-210 as a normalization control. It is known that miR-210 is significantly affected by RBC hemolysis, as well as miR-16. However, unlike miR-16, miR-210 is not abundant in platelets [28,40]. The hemolysis assessment and medication therapies were not evaluated in the test sample of the referred study [22]. Similar levels of miR-16, miR-451a, and miR-210 between ACM patients and controls in the test sample indicate the lack of differences in RBC hemolysis. However, the increase in the main platelet-abundant miRNAs (miR-223: 6.3-fold, miR-126: 2.3-fold, miR-21: 2.4-fold, and some others) in ACM vs. controls suggests platelet or PMP-derived changes in the miRNA signature in ACM. The decrease in the relative miR-320a level apparently occurred against the backdrop of platelet activation and PMP release. Some reasonable questions arise: Is the miRNA profile a marker of disease or a change in the platelet state due to a specific type of therapy? How meaningful is it to assess differences in the miRNA profile without considering the type of therapy?

Previously, it was shown that antiplatelet drugs and anticoagulants have the ability to alter the circulating extracellular plasma miRNA profile [18,21]. As miRNAs are considered to be biomarkers for platelet activation, the issue of a relationship between medication therapies and circulating miRNAs is relevant to any other therapy potentially influencing platelet activation, including ACEI beta-blockers, calcium channel blockers, and statins. In our study we analyzed the influence of these medication therapies to target miRNA and found a relationship between ACEI treatment and plasma levels of miR-320a-3p, miR-146a-5p, and miR-21-5p. A significant increase in miR-320a-3p plasma levels in hypertensive and PAF patients receiving ACEI leads to speculation as to whether this miRNA is a marker of disease or a result of ACEI therapy.

The reason for the increase in plasma miR-320a-3p in PAF patients in our study cannot be unambiguously assessed. Since miR-320a is one of the 20 most abundant miRNAs in plasma exosomes [41], its increase may be exosomal in origin. In the study by Wang et al. [11], miR-320a was the most abundant among the miRNAs that exhibited significantly elevated relative levels in plasma exosomes in patients with persistent AF compared to those with a sinus rhythm. The data on the exosome miRNA profile were obtained by small RNA sequencing, but the elevated levels of plasma exosomal miR-320a were not confirmed by qPCR with small nuclear RNA (snRNA) U6 as a reference RNA in an extended cohort. The use of U6 as a normalization control for exosomal miRNA qPCR analysis is questionable due to the different size of snRNAs compared to miRNA and the low presence of snRNAs in exosomes [41]. For total plasma/serum miRNA studies, U6 is not suitable as a reference [42]. The miRNA compositions of PMP and exosomes differ in the most abundant miRNAs, and the overrepresentation of certain microparticle fractions in plasma can result in a bias in the overall miRNA composition. The observed increase in miR-320a levels in patients receiving ACEI therapy provides a key to understanding this effect. ACEI therapy is known to reduce platelet activity and the ability of platelets to aggregate [43]. Decreased platelet activity may be the cause of reduced PMP release [44,45]. Therefore, a decrease in PMP-derived miRNAs can increase the presence of other miRNA fractions, e.g., exosome-derived factors, in the general circulating RNA pool.

In our work, we also found that the relative plasma levels of miR-320a-3p, as well as miR-146a-5p and miR-21-5p, were positively correlated with the CHA_2_DS_2_-Vasc score related to the stroke risk. The miR-146a-5p miRNA is cytokine-responsive miRNA that regulates distinct components of NF-κB signaling and is known to be involved in atherogenesis as it controls endothelial activation and dysfunction [46]. The relative plasma miR-146a-5p level is elevated in heart failure patients and positively correlates with coronary heart disease severity [47,48]. In advanced atherosclerosis, miR-21-5p is also considered to be a key modulator of pathologic processes and its overexpression promotes the stabilization of vulnerable plaques [49]. A number of studies indicate that an elevated miR-21-5p circulating level might be used as a predictor for heart failure and CAD [50,51]. In addition to miR-146a-5p and miR-21-5p, miR-320a-3p also contributes to atherogenesis and its plasma level is higher in CAD patients [52]. Considering all of these facts, we can speculate that the observed association of these three miRNAs with the CHA_2_DS_2_-Vasc score may be related to atherosclerosis progression, since the parameters used for the CHA_2_DS_2_-Vasc calculation (congestive heart failure, hypertension, ageing, diabetes, stroke, and vascular state) are closely linked to atherosclerosis.

Several important study limitations should be noted. First, a considerable age bias existed in the compared groups: PAF patents were ~10 years older than hypertensive patients and ~20 years older than healthy controls. This bias came from the initial study design: the formation of a sample based on outpatient cardiology clinic patients with the described inclusion and exclusion criteria naturally resulted in an age bias between controls and PAF patients. Epidemiological studies indicate that in North America and Europe, the sex–age-specific incidence of AF increases progressively after 60 years, before peaking at around 80 years [34]. Therefore, the present study cannot be considered as properly case-control age- and sex-matched. To estimate the age contribution to the obtained results, we performed two additional variants of MLR analysis for a comparison of miRNA relative plasma levels in the study sample groups. First, we included age as an additional confounding factor in the MLR analysis of the comparison of miRNA relative plasma levels in the study sample groups. The significant differences with *p* < 0.05 persisted for the PAF vs. CONTR comparison for miR-146a-5p, miR-19a-3p, and miR-320a-3p. We further performed MLR analysis of the relative target miRNA plasma levels in order to obtain *p*-values of significance for age prediction without any clinical parameters as confounding factors. This analysis showed that miR-146a-5p, miR-21-5p, and miR-320a-3p plasma levels have a significant (*p* < 0.05) relationship with age. Given that these three miRNAs are related to atherogenesis and in our study were associated with the CHA_2_DS_2_-Vasc score, we suggest that the relationship with age is probably a result of targeted group formation based on the specific clinical parameters.

Furthermore, only one endogenous reference (hsa-miR-16-5p) was used for normalization of the miRNA expression data and the relative plasma level for each analyzed miRNA was calculated using this reference. Given the existing data on the influence of pre-analytical factors and complex heterogeneity of circulating miRNA origin, the analysis of miRNA signatures is more informative than the changes in individual miRNAs, and using at least three different reference miRNAs is preferable. Small RNA sequencing with further qPCR validation is likely to be the optimal approach for studying circulating extracellular blood miRNAs.

Another limitation of this study is the use of a single technical replicate of each miRNA qPCR analysis in each sample. However, we showed the technical coefficients of variation (CV) of miRNA qPCR measurements, and aimed to use an adequate number of biological replicates. In this study, we did not analyze miRNA species abundant in platelets and PMPs (miR-223 and miR-126). As changes in the miRNA profile may occur due to platelet activation or its inhibition by medical therapies, we recommend including these miRNAs in the qPCR analysis of plasma miRNA in any studies on CVD patients.

In conclusion, despite a number of limitations, this study was carried out in accordance with the main current recommendations for circulating miRNA research and reveals the utility of previously known AF-associated circulating miRNAs as potential biomarkers for PAF. Using data on the influence of pre-analytical and analytical conditions, the presence of concomitant diseases, the use of medication therapies, and the biological role and origin of extracellular circulating miRNAs, we investigated and discussed the potential background for the observed miRNA plasma level changes in PAF patients.

## 4. Materials and Methods

### 4.1. Ethics Statement

The study complied with the Declaration of Helsinki, and the study protocol was approved by the Ethics Committee at First Moscow State Medical University (Sechenov University), Moscow, Russia, on 14 July 2017 (application no. 05-17). Written informed consent was obtained from all study participants.

### 4.2. Study Population

Outpatient adult cardiology clinic patients (males and females from 40 to 90 years old) were included in the study. Several patient groups were recruited: patients with known PAF (including those with cardiovascular comorbidities not in the exclusion criteria listed below); hypertensive patients without AF; and healthy controls. AF was diagnosed by the typical electrocardiographic signs of this arrhythmia on a resting ECG and/or 24-h ECG monitoring. PAF was determined in accordance with the 2016 ESC recommendations. HT was diagnosed by the daily monitoring of blood pressure and/or antihypertensive therapy. CAD was verified by coronary angiography. The exclusion criteria were clinical heart failure, a reduced left ventricular ejection fraction (<45%), acute coronary syndrome within the previous three months, percutaneous coronary intervention within the previous three months, heart valve surgery, pacemaker implantation, congenital heart disease, arterial pulmonary hypertension, inflammatory cardiac diseases, cardiomyopathy, arrhythmia ablation within the previous three months, severe chronic kidney disease (GFR < 45 mL/min/1.73 sq.m), primary kidney and liver diseases, severe chronic obstructive pulmonary disease, rheumatic and autoinflammatory diseases, active cancer, haematological diseases, neurology or psychiatry diseases, any major surgery within the previous three months, pregnancy and lactation, and acute medical illness within 30 days prior to inclusion and inability to provide informed consent. In the PAF group, cardioverted patients were included for seven days or more after the procedure.

### 4.3. Plasma Collection and Storage

Whole blood was collected in 6 mL K2-EDTA tubes (BD, Franklin Lakes, NJ, USA) and stored at room temperature for no more than one hour prior to plasma isolation. To obtain plasma for miRNA isolation, the whole blood sample was processed in two steps: whole blood was centrifuged at 2130× *g* for 10 min at room temperature (Step 1), and the crude plasma was then repeatedly centrifuged in sterile 15 mL conical tubes at 2130× *g* for 10 min at room temperature. The upper two-thirds of the plasma layer was stored at −20 °C in 500 μL portions in sterile RNAse-free 1.5 mL tubes in the clinical facility. Within 1 month, frozen plasma samples were transported to the laboratory facility without thawing and then stored at −80 °C. The isolation of miRNA was performed within one month after blood collection. Before miRNA isolation, plasma samples were thawed on ice and centrifuged at 16,000× *g* for 15 min at 4 °C to pellet down any residual cell debris (Step 3). The supernatant in a volume of 300 µL was immediately used for miRNA extraction and 10 µL was aliquoted and stored at −20 °C for further hemolysis assessment.

### 4.4. Hemolysis Assessment of Plasma Samples

Only samples without visually detected hemolysis were included in this study. Low-level RBC hemolysis in plasma/serum samples was assessed by the spectrophotometric measurement of absorbance at a 414 nm wavelength (peak of free hemoglobin). For each sample, a 10 µL aliquot of plasma supernatant from centrifugation Step 3 was used for hemolysis assessment within seven days after miRNA isolation. The sample was thawed, incubated at room temperature for 30 min, and analyzed on a NanoDrop^®^ 2000 spectrophotometer (Thermo Fisher Scientific, Waltham, MA, USA) by the measurement of ultraviolet-visible (UV-Vis) absorbance with a 1 mm path at 385 nm (A385) and 414 nm (A414) wavelengths in triplicate for each sample. For each measurement, the following indices were calculated based on the mean A414 and A385 values: ∆(A414 − A385), a lipemia-independent hemolysis score, HS = ∆(A414 − A385) + 0.16 × A385, and the A414/A385 ratio [35]. Samples with HS > 0.25 were not included in this study.

### 4.5. Plasma miRNA Isolation

miRNA was isolated from 300 µL of the plasma supernatant from centrifugation Step 3 using a NucleoSpin miRNA Plasma kit (Macherey-Nagel, Düren, Germany) according to the manufacturer’s guidelines. Proteinase K digestion was performed for each plasma supernatant sample before the isolation of miRNA. Each miRNA sample had a total volume of 30 µL and was stored at −80 °C prior to cDNA synthesis.

### 4.6. cDNA Synthesis and qPCR for miRNA Detection

A 2-µL sample of miRNA was used for cDNA synthesis with a TaqMan Advanced miRNA cDNA Synthesis Kit (Thermo Fisher Scientific, Waltham, MA, USA), according to the manufacturer’s recommendations, using DNA Engine Tetrad 2 Thermal Cycler (Bio-Rad Laboratories, Hercules, CA, USA). Twelve commercially available TaqMan Advanced miRNA assays with TaqMan Fast Advanced Master Mix (Thermo Fisher Scientific, Waltham, MA, USA) were used to perform qPCR, according to the manufacturer’s protocol. The list of miRNA assays with catalogue numbers and mature miRNA sequences is given in Table 5. The list includes the 10 main miRNAs potentially involved in AF pathogenesis (expressed in the heart or associated with atrial remodeling or fibrillation), with previously reported altered plasma levels in AF patients [6]. As an endogenous normalization control, hsa-miR-16-5p was chosen, since it is widely used for plasma miRNA studies. For qPCR-based hemolysis assessment, a pair of miRNAs (hsa-miR-23a-3p and hsa-miR-451a) that indicate the degree of hemolysis by their ratio were included in the study [19]. The majority of analyzed miRNAs (excluding miR-409-3p and hsa-miR-432-5p) are amongst the most commonly found circulating miRNAs in plasma [19]. For each miRNA assay, a no-template control (NTC) containing nuclease-free water instead of miRNA sample was analyzed. We performed qPCR using the QuantStudio 5 Real-Time PCR system (Thermo Fisher Scientific, Waltham, MA, USA) in MicroAmp 96-well PCR plates and optical adhesive film, in a “Fast” cycling mode with the following program: enzyme activation: 20 s at 95 °C; 45 cycles: denature for 1 s at 95 °C, and anneal/extend for 20 s at 60 °C. We obtained qPCR data using QuantStudio Design and Analysis Software v1.4.1 (Thermo Fisher Scientific, Waltham, MA, USA). Cq values were calculated using the automatic “Baseline” value and the experimentally set “Threshold” value of ∆Rn = 0.15 for all analyzed miRNA targets. Cq measurements were performed in a single technical replicate for each miRNA target within an individual sample. Normalization of qPCR data for each target miRNA was performed using the Cq value of hsa-miR-16-5p. For each miRNA analyzed, its relative plasma expression level was calculated as 2^−ΔCq (target miRNA–hsa-miR-16-5p)^. If the Cq value of the analyzed miRNA was undetectable by 45 qPCR cycles, its Cq was considered to be 45. The impact of RBC hemolysis on circulating miRNAs was estimated by the qPCR-detected ratio of hemolysis-dependent miRNA miR-451a abundant in RBC and miR-23a-3p, which is hemolysis-independent [19]. The difference between Cq values of these miRNAs was calculated by the formula dCq _(miR-23a-3p–miR-451a)_ = Cq(hsa-miR-23a-3p) − Cq(hsa-miR-451a). For all the samples in this study, the same laboratory workflow and PCR data analysis protocol was used.

In our test experiments with plasma miRNA samples, the Cq values of hsa-miR-432-5p were very low (typical Cq 35 to 45), with ∆Rn values at the 45th PCR cycle that were ~10-fold lower than the lowest expressed miRNA hsa-miR-409-3p and ~30-fold lower than hsa-miR-16-5p. Therefore, hsa-miR-432-5p was excluded from further analysis.

To estimate the qPCR performance for miRNA detection, we obtained miRNA from two random independent plasma samples. Prior to cDNA synthesis, miRNA samples were spiked-in with synthetic miRNA cel-miR-39-3p in final concentrations of 1 and 10 pM. We performed qPCR for three technical replicates for all analyzed miRNAs, as well as for spike-in miRNA (478293_mir TaqMan Advanced assay for cel-miR-39-3p). The mean Cq values for each of the analyzed miRNAs are given in Appendix A. Cq values of the NTC for all analyzed miRNAs were undetectable at the 45th cycle of qPCR. Coefficients of variation (CV) for the relative expression level for each analyzed miRNA were calculated for the three qPCR technical replicates for two independent samples. The maximum CV values did not exceed 12% (Appendix A). Spike-in miRNA was only used as a control for cDNA synthesis in the qPCR performance assessment step.

### 4.7. Data Analysis

To determine statistically significant differences in relative plasma levels of the analyzed miRNAs between the study groups, *p*-values were calculated using MLR with Bonferroni–Holm correction for multiple comparisons. Since it had been previously shown that hemolysis affects the plasma levels of some miRNAs, we used quantitative values that reflected the hemolysis degree—HS and dCq _(miR-23a-3p–miR-451a)_—as confounding factors in each MLR analysis performed in this study. Since some of the miRNAs included in the study have altered plasma levels in patients with DM and CAD, we added the presence of type 2 DM and the presence of stable CAD as confounding factors in the MLR analysis of study groups [2,53]. Differences with *p*-value < 0.05 were considered statistically significant.

## Figures and Tables

**Figure 1 ijms-21-03485-f001:**
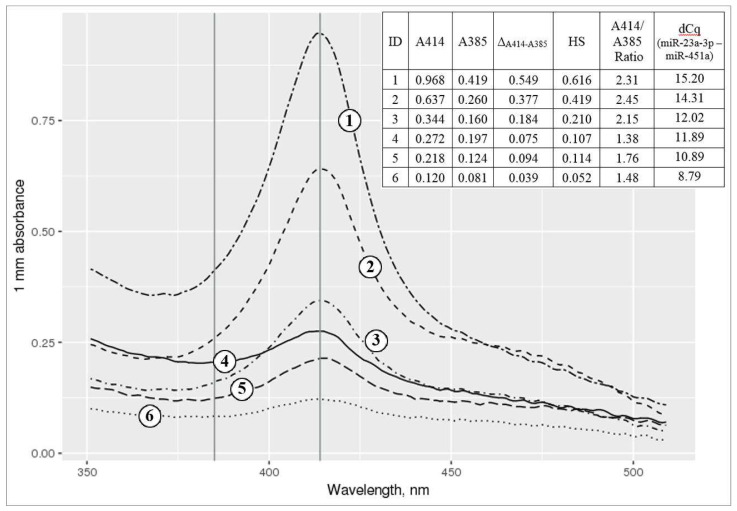
Spectrophotometric analysis of six different plasma supernatant samples in the 350–500 nm wavelength range. Left and right vertical lines correspond to 385 and 414 nm wavelengths. 1–2: hemolyzed samples with (1) and without (2) visually detected hemolysis, not included in the study; 3–6: samples with a low hemolysis degree, included in the study. Spectrophotometric data are supplemented by miRNA qPCR-based hemolysis assessment data.

**Figure 2 ijms-21-03485-f002:**
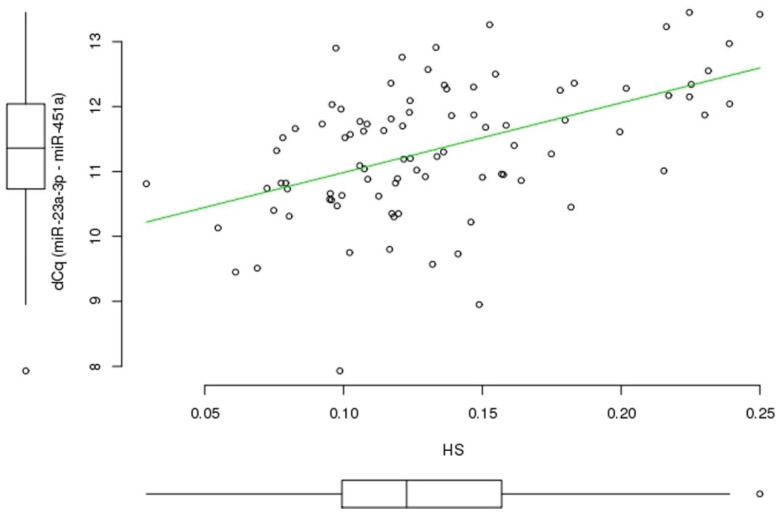
dCq _(miR-23a-3p–miR-451a)_ and HS values in plasma samples of 90 study participants. HS: hemolysis score; dCq (miR-23a-3p–miR-451a): Cq difference between miR-23a-3p and miR-451a. The boxplots near the x- and y-axis represent the median and interquartile ranges (IQR) in the box, minimum and maximum values in the “whiskers”, and outliers in the dots.

**Figure 3 ijms-21-03485-f003:**
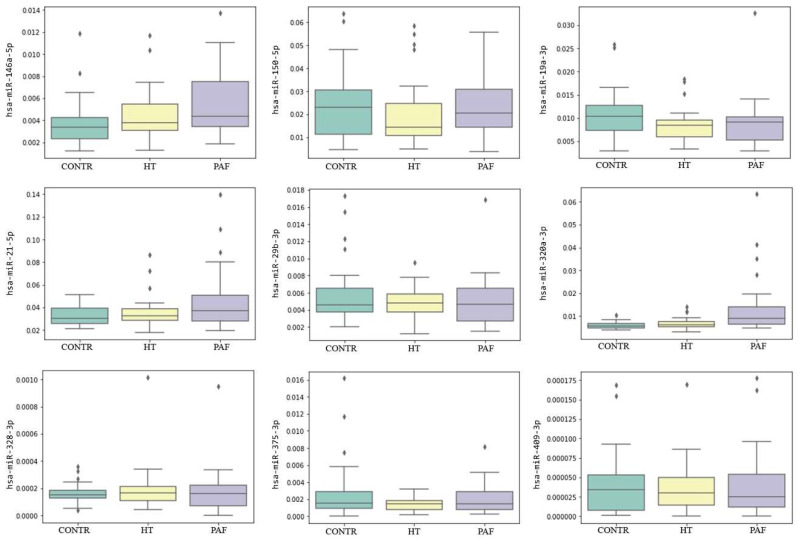
Relative miRNA plasma levels in the study sample groups: PAF: 30 PAF patients; HT: 30 hypertensive patients without AF; CONTR: 30 healthy controls. On the y-axis, the relative miRNA plasma levels (normalized to miR-16-5p) are shown. The boxplots represent the median and interquartile ranges (IQR) in the box, minimum and maximum values in the “whiskers”, and outliers in the rhombic dots.

**Figure 4 ijms-21-03485-f004:**
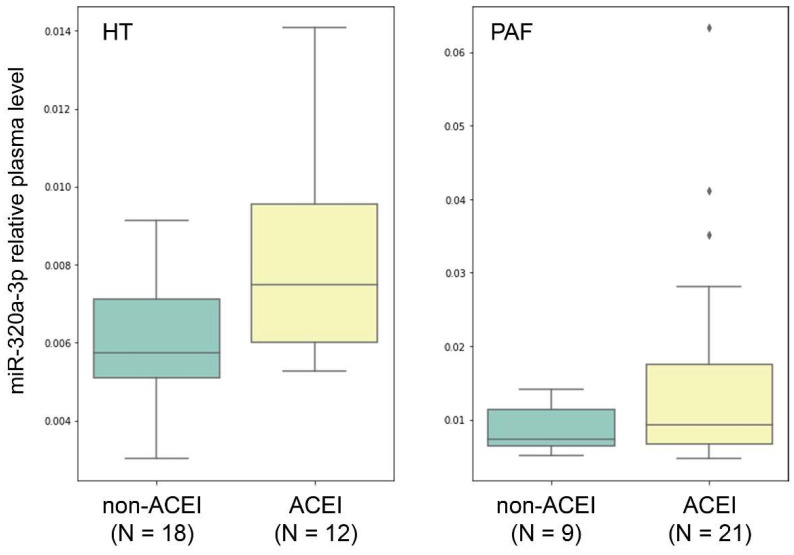
Distribution of miR-320a-3p relative plasma levels in the study sample groups HT (hypertensive patients without AF, N = 30) and PAF (patients with paroxysmal atrial fibrillation) in patients with and without angiotensin-converting enzyme inhibitors (ACEI) treatment. The boxplots represent the median and interquartile ranges (IQR) in the box, minimum and maximum values in the “whiskers”, and outliers in the rhombic dots.

**Figure 5 ijms-21-03485-f005:**
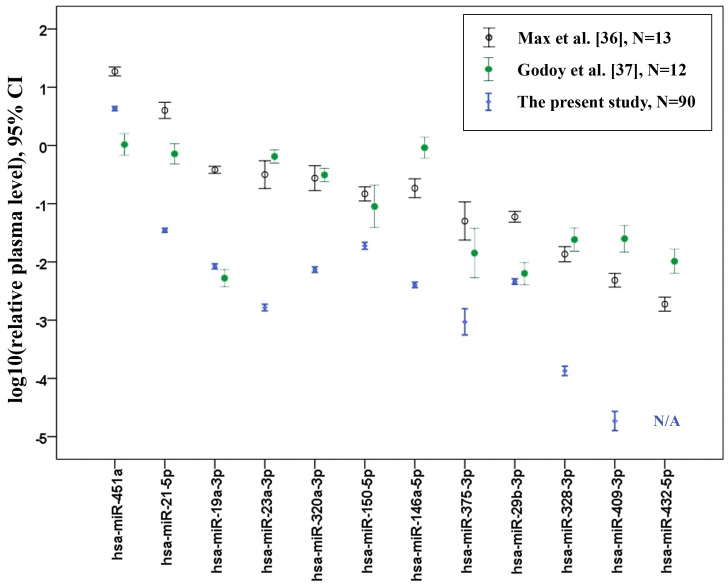
Comparison of miR-16-5p-normalized relative plasma levels between the present study and the two most recent miRNA plasma sequencing studies performed on healthy adults. Error bars represent the mean and 95% confidence interval (CI). N/A: not applicable.

**Table 1 ijms-21-03485-t001:** Characteristics of the study sample groups: PAF: 30 paroxysmal atrial fibrillation (PAF) patients; HT: 30 hypertensive patients without atrial fibrillation (AF); CONTR: 30 healthy controls. SD: standard deviation; ACEI: angiotensin-converting enzyme inhibitors.

Group ID	CONTR	HT	PAF	Total
Number of patients	30	30	30	90
Mean age (SD), years	47.3 (5.6)	57.7 (9.5)	67.6 (10.0)	57.5 (11.9)
Gender (male/female)	15/15	17/13	15/15	47/43
Type 2 DM presence	0	0	5	5
Stable CAD presence	0	0	7	7
HT presence	0	30	30	60
Left atrial volume, mL, mean (SD)	49.9 (6.2)	54 (8.5)	79.7 (26.2)	61.2 (20.9)
CHA_2_DS_2_-Vasc, mean (SD)	0.00 (0.00)	1.77 (0.94)	2.87 (1.31)	1.54 (1.50)
**Blood Lipid Profile**				
Total cholesterol, mmol/L, mean (SD)	4.55 (0.90)	5.19 (0.98)	4.85 (1.40)	4.86 (1.13)
Triglycerides, mmol/L, mean (SD)	1.13 (0.31)	1.46 (0.50)	1.52 (0.88)	1.37 (0.63)
LDL cholesterol, mmol/L, mean (SD)	2.14 (0.56)	2.69 (0.67)	2.48 (1.15)	2.44 (0.85)
HDL cholesterol, mmol/L, mean (SD)	1.86 (0.51)	1.84 (0.50)	1.65 (0.52)	1.78 (0.51)
**Medication Therapies**				
Beta-blockers	2	15	15	32
Calcium channel blockers	0	4	14	18
ACEI	0	12	21	33
Diuretics	0	4	13	17
Antiplatelet drugs	0	9	3	12
Anticoagulants	0	3	22	25
Statins	1	11	13	25

**Table 2 ijms-21-03485-t002:** Characteristics of hemolysis assessment in the study sample groups: PAF: 30 PAF patients; HT: 30 hypertensive patients without AF; CONTR: 30 healthy controls. A414 and A385: spectrophotometric absorbance at 414 and 385 nm wavelengths obtained during hemolysis assessment, respectively; HS: hemolysis score; dCq _(miR-23a-3p–miR-451a)_: Cq difference between miR-23a-3p and miR-451a.

Group ID	CONTR	HT	PAF	Total
A414	0.225 (0.066)	0.288 (0.076)	0.259 (0.077)	0.257 (0.077)
∆(A414-A385)	0.102 (0.042)	0.117 (0.053)	0.111 (0.041)	0.110 (0.046)
HS	0.122 (0.045)	0.144 (0.053)	0.134 (0.044)	0.134 (0.048)
A414/A385 Ratio	1.86 (0.30)	1.76 (0.37)	1.78 (0.24)	1.80 (0.31)
dCq _(miR-23a-3p–miR-451a)_	11.10 (0.98)	11.56 (1.07)	11.38 (1.01)	11.34 (1.03)

**Table 3 ijms-21-03485-t003:** Multiple linear regression (MLR) *p*-values for a comparison of miRNA relative plasma levels in the study sample groups: PAF: 30 PAF patients; HT: 30 hypertensive patients without AF; CONTR: 30 healthy controls. The asterisks (*) indicate statistically significant differences (*p* < 0.05). For each significant change between groups, the direction of change (up/down) and log2(fold change) values are given.

Group Comparison	miRNA								
	miR-146a-5p	miR-150-5p	miR-19a-3p	miR-21-5p	miR-29b-3p	miR-320a-3p	miR-328-3p	miR-375-3p	miR-409-3p
PAF vs. CONTR	0.012 * up 0.533	0.285	0.026 * down −0.294	0.248	0.068	0.000 * up 1.195	0.248	0.454	0.248
PAF vs. HT	0.667	0.667	0.667	0.912	0.732	0.020 * up 0.977	0.667	0.667	0.986
CONTR vs. HT	0.075	0.636	0.376	0.376	0.608	0.382	0.690	0.608	0.388

**Table 4 ijms-21-03485-t004:** Association between relative miRNA plasma levels and the CHA_2_DS_2_-Vasc score in the study sample.

Type of Analysis	Spearman’s Correlation Analysis		MLR Analysis: CHA2DS2-Vasc ≥ 2 vs. CHA2DS2-Vasc < 2	
miRNA	Rho Correlation Coefficient	*p*-Value	log2(Fold Change)	*p*-Value
miR-146a-5p	0.551	1.64 × 10^−11^	1.574	2.57 × 10^−4^
miR-150-5p	0.238	0.007	1.395	0.972
miR-19a-3p	−0.093	0.297	−0.208	0.647
miR-21-5p	0.521	2.88 × 10^−10^	1.585	1.56 × 10^−4^
miR-29b-3p	0.243	0.006	0.716	0.846
miR-320a-3p	0.575	1.25 × 10^−12^	1.680	1.03 × 10^−5^
miR-328-3p	0.221	0.012	0.810	0.670
miR-375-3p	0.172	0.052	1.344	0.670
miR-409-3p	0.138	0.121	1.131	0.972

**Table 5 ijms-21-03485-t005:** List of the miRNA assays used for qPCR. Assay IDs are given for TaqMan Advanced miRNA assays (Thermo Fisher Scientific, Waltham, MA, USA).

Assay Name	Assay ID	Mature miRNA Sequence	Type of miRNA
hsa-miR-16-5p	477860_mir	UAGCAGCACGUAAAUAUUGGCG	Normalization control
hsa-miR-23a-3p	478532_mir	AUCACAUUGCCAGGGAUUUCC	Hemolysis assessment
hsa-miR-451a	478107_mir	AAACCGUUACCAUUACUGAGUU	Hemolysis assessment
hsa-miR-146a-5p	478399_mir	UGAGAACUGAAUUCCAUGGGUU	Candidate for AF
hsa-miR-150-5p	477918_mir	UCUCCCAACCCUUGUACCAGUG	Candidate for AF
hsa-miR-19a-3p	479228_mir	UGUGCAAAUCUAUGCAAAACUGA	Candidate for AF
hsa-miR-21-5p	477975_mir	UAGCUUAUCAGACUGAUGUUGA	Candidate for AF
hsa-miR-29b-3p	478369_mir	UAGCACCAUUUGAAAUCAGUGUU	Candidate for AF
hsa-miR-320a-3p	478594_mir	AAAAGCUGGGUUGAGAGGGCGA	Candidate for AF
hsa-miR-328-3p	478026_mir	CUGGCCCUCUCUGCCCUUCCGU	Candidate for AF
hsa-miR-375-3p	478074_mir	UUUGUUCGUUCGGCUCGCGUGA	Candidate for AF
hsa-miR-409-3p	478084_mir	GAAUGUUGCUCGGUGAACCCCU	Candidate for AF
hsa-miR-432-5p	478101_mir	UCUUGGAGUAGGUCAUUGGGUGG	Candidate for AF

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
