# Peer review of "Elevated Plasma Levels of Circulating Extracellular miR-320a-3p in Patients with Paroxysmal Atrial Fibrillation"

_ijms, 2020, doi:10.3390/ijms21103485_

Round 1

Reviewer 1 Report

In this manuscript, the authors demonstrated that circulating miR-320a-3p may be considered as potential paroxysmal atrial fibrillation (PAF) biomarkers. This is a well-written manuscript. Although the study is important, the manuscript needs major improvements as summarized in my comments below.

Major comments:

  1. Why the authors selected miR-146a-5p, miR-150-5p, miR-21-5p, miR-29b-3p, miR-320a-3p, miR-328-3p, miR-375-3p and miR-409-3p for further investigation? Did any novel miRNAs associate with PAF?
  2. I suggest that 2.2 sections can move to the material and method. It looks like methodology.
  3. The case numbers used in the figures should be shown in the figures.
  4. The y-axis of figure 3 and figure 4 should be defined. Please add more descriptions in these figure legends.
  5. Did any combination effect of miRNAs associated with PAF or ACEI?

     6. The discussion is too short. Please add more comments based on the          results. The final conclusion should be described in the last section.

Reviewer 2 Report

This study raises an important issue of novel non-invasive biomarkers of atrial fibrillation. The study design is logical, groups are divided at minimum to reach statistical significance or correlation reliability. The methodology might be also considered as reliable. On the other hand, authors try to incorporate some large unit about the problem of hemolysis, which also very important, however the link of biomarker and hemolysis is weakly explained.   

Manuscript brings some data worth to be published, however there are some minor and major problems which in my opinion need to be improved, corrected or discussed.

1. Authors should reconsider using AF and PAF abbreviations because it’s might be confusing. Once in the text is written PAF, whereas in tables AF, and so on. I’m sure that it’s well known for the authors that Paroxysmal atrial fibrillation (PAF) constitutes approximately half of all atrial fibrillation (AF) cases and is thought to represent an early stage of the disease. However, it’s might be not so obvious for all potential readers.

2. I believe the table 2. with characteristics of qPCR detection is unnecessary (could easily transfer to supplementary data). These results could be described in a few sentences, I recommend removing or justify why is crucial for study.

3. In my opinion table 4. “MLR p-values for comparison of miRNA relative plasma levels” is unclear. Firstly MLR is, I assume, multiple linear regression, however it’s explain only in abbreviations at the end of the manuscript. Secondly, description is needed, what are stars I assume they intended to flag levels of significance.   

4. Furthermore, figure 3. lacks a description of what are bars, what are dots. We can only assume that those are: min max or IQR.

5. Supplementary table lack of description, it’s separate file this should be added (HT, PAF…).

6. Further, the major problem for received results is the fact, that controls were 10 years younger than HT and 20 years younger than PAF. This is an important issue and needs to be discussed to a unique accusation of bias. Furthermore 1 year SD is doubtfully low. But, it might be just mistake or intended procedure?   

Some issues should be discuss or analyzed despite limited number of patients in subgroup such as medication therapies, antiplatelet drugs etc. which were shown in characteristics of the study sample groups.

Discussion is short however it might be easily reconstructed and improved basing just on sentences for methods or introduction. The purpose of the discussion is to interpret and describe the significance of your findings.

Finally, the text should be checked for correctness of grammar and commas, and some phrases are “wordy”.

For example:

in line 45 – “their structural, metabolic and electrophysiologic changes is an essential component ” –“changes are, however change is”

line 85 - In the current study we attempted to fulfill… I’m uncertain but the coma is missing…

line 155 “The MLR analysis showed that 3 miRNAs “ please use text instead of numbers.

and so on…

Round 2

Reviewer 1 Report

The authors have satisfactorily responded to all my questions.

Reviewer 2 Report

I appreciate the authors' effort in response to my review. The manuscript have been significantly improved. I believe it might be accepted in present form.